# Nonlinear Acceleration of CNNs

**Damien Scieur**[\*], **Edouard Oyallon**[†], **Alexandre d'Aspremont**[\*] **and Francis Bach**[\*]
[\*]DI, Ecole Normale Supérieure, UMR CNRS 8548, INRIA, PSL Research University.
[†]CVN, CentraleSupélec, Université Paris-Saclay; Galen team, INRIA Saclay

## Abstract

Regularized Nonlinear Acceleration (RNA) can improve the rate of convergence of many optimization schemes such as gradient descent, SAGA or SVRG, estimating the optimum using a nonlinear average of past iterates. Until now, its analysis was limited to convex problems, but empirical observations show that RNA may be extended to a broader setting. Here, we investigate the benefits of nonlinear acceleration when applied to the training of neural networks, in particular for the task of image recognition on the *CIFAR10* and *ImageNet* data sets. In our experiments, with minimal modifications to existing frameworks, RNA speeds up convergence and improves testing error on standard CNNs.

## 1 Introduction

Successful deep Convolutional Neural Networks (CNNs) for large-scale classification are typically optimized through a variant of the stochastic gradient descent (SGD) algorithm (Krizhevsky et al., 2012). Refining this optimization scheme is a complicated task and requires a significant amount of engineering whose mathematical foundations are not well understood (Wilson et al., 2017). Here, we propose to wrap an adhoc acceleration technique known as Regularized Nonlinear Acceleration algorithm (RNA) (Scieur et al., 2016), around existing CNN training frameworks. RNA is generic as it does not depend on the optimization algorithm, but simply requires several successive iterates of gradient based methods, which involves a minimal adaptation in many frameworks. This meta-algorithm has been applied successfully to gradient descent in the smooth and strongly convex cases, with convergence and rate guarantees recently derived in (Scieur et al., 2016). Recent works (Scieur et al., 2017) further show that it improves standard stochastic optimization schemes such as SAGA or SVRG (Defazio et al., 2014; Johnson & Zhang, 2013), which indicates it may also be a strong candidate as an accelerated method in stochastic non convex cases.

RNA is an ideal meta-learning algorithm for deep CNNs, because contrary to many acceleration methods Lin et al. (2015); Güler (1992), optimization can be performed off-line and does not involve any potentially expensive extra-learning process. This means one can focus on acceleration *a posteriori*. RNA related numerical computations are not expensive, and form a simple linear system from a well-chosen linear combination of several optimization steps. This system is usually very small relative to the number of parameters, so the cost of acceleration grows linearly with respect to this number.

Here, we study applications applications of RNA to several recent architectures, like ResNet (He et al., 2016), applied to classical challenging datasets, like CIFAR10 or ImageNet. Our contributions are are twofold: first we demonstrate that it is often possible to achieve an accuracy similar to the final epoch in half the time; second, we show that RNA slightly improves the test classification performance, at no additional numerical cost. We provide an implementation that can be incorporated using only few lines of code around many standard Python deep learning frameworks, like PyTorch[1].

---

[1]Code can be found here:
`https://github.com/windows7lover/RegularizedNonlinearAcceleration`

## 2  ACCELERATING WITH REGULARIZED NONLINEAR ACCELERATION

This section briefly describes the RNA procedure and we refer the reader to Scieur et al. (2016) for more extensive explanations and theoretical guarantees. For the sake of simplicity, we consider an iterate sequence $\{\theta_k\}_{0 \le k \le K}$ of $K+1$ elements of $\mathbb{R}^d$ produced from the successive steps of an iterative optimization algorithm. For example, each $\theta_k$ could correspond to the parameters of a neural network at epoch $i$, trained via a gradient descent algorithm, i.e.

$$\theta_{k+1} = \theta_k - \eta \nabla f(\theta_k),$$

with $\eta$ the step size (or learning rate) of the algorithm. Local minimization of $f$ is naturally achieved by $\theta^*$ where $\nabla f(\theta^*) = 0$. RNA aims to linearly combine the parameters $\theta_k$ into an estimate $\hat{\theta}$

$$\hat{\theta} = \sum_{k \le K} c_k \theta_k \text{ s.t. } \sum_{k \le K} c_k = 1, \tag{1}$$

so that $\nabla f(\hat{\theta})$ becomes smaller. In other terms, RNA output $\hat{\theta}$ which solves approximately

$$\min_c \left\| \nabla f \left( \sum_{k \le K} c_k \theta_k \right) \right\|^2 \quad \text{subject to } \sum_{k \le K} c_k = 1. \tag{2}$$

In the next subsection, we describe the algorithm and intuitively explain the acceleration mechanism when using the optimization method (1), because this restrictive setting makes the analysis simpler.

### 2.1  REGULARIZED NONLINEAR ACCELERATION ALGORITHM

In practice, solving (2) is a difficult task. Instead, we will assume that the function $f$ is *approximately* quadratic in the neighbourhood of $\{\theta_k\}_{k \le K}$. This approximation is common in optimization for the design of (quasi-)second order methods, such as the Newton's method or BFGS. Thus, $\nabla f$ can be considered almost as an affine function, which means:

$$-\nabla f \left( \sum_{k \le K} c_k \theta_k \right) \approx \sum_{k \le K} c_k \nabla f(\theta_k). \tag{3}$$

From a finite difference scheme, one can easily recover $\{\nabla f(\theta_k)\}_k$ from the iterates in (1), because for any $k$, we have $-\eta \nabla f(\theta_k) = (\theta_{k+1} - \theta_k)$. As linearized iterates of a flow tends to be aligned, minimizing the $\ell^2$-norm of (3) requires incorporating some regularization to avoid ill-conditioning

$$\min_c \|Rc\|^2 + \lambda \|c\|^2 \quad \text{subject to } \sum_{k \le K} c_k = 1,$$

where $R = [\theta_1 - \theta_0, \ldots, \theta_K - \theta_{K-1}]$. This exactly corresponds to the combination of steps 2 and 3 of Algorithm 1. Similar ideas hold is the stochastic case (Scieur et al., 2017), under limited assumption on the signal to noise ratio.

---

**Algorithm 1** Regularized Nonlinear Acceleration (**RNA**), (and computational complexity).

---

**Input:** Sequence of vectors $\{\theta_0, \theta_1, \ldots, \theta_K\} \in \mathbb{R}^d$, regularization parameter $\lambda > 0$.
  1: Compute $R = [\theta_1 - \theta_0, \ldots, \theta_K - \theta_{K-1}]$                                        $O(K)$
  2: Solve $(R^T R + \lambda I)z = \mathbf{1}$.                                                                  $O(K^2 d + K^3)$
  3: Normalize $c = z/(\sum_{k \le K} z_k)$.                                                                     $O(K)$
**Output:** $\hat{\theta} = \sum_{k \le K} c_k \theta_k$.                                                        $O(Kd)$

---

### 2.2  PRACTICAL USAGE

We produced a software package based on PyTorch that includes in minimal modifications of existing standard procedures. As claimed, the RNA procedure does **not** require any access to the data, but simply stores regularly some model parameters in a buffer. On the fly acceleration on CPU is achievable, since one step of RNA is roughly equivalent to squaring a matrix of size $d \times K$, to form a $K \times K$ matrix and solve the corresponding system. $K$ is typically 10 in the experiments that follow.

## 3  APPLICATIONS TO CNNS

We now describe the performance of our method on classical CNN training problems.

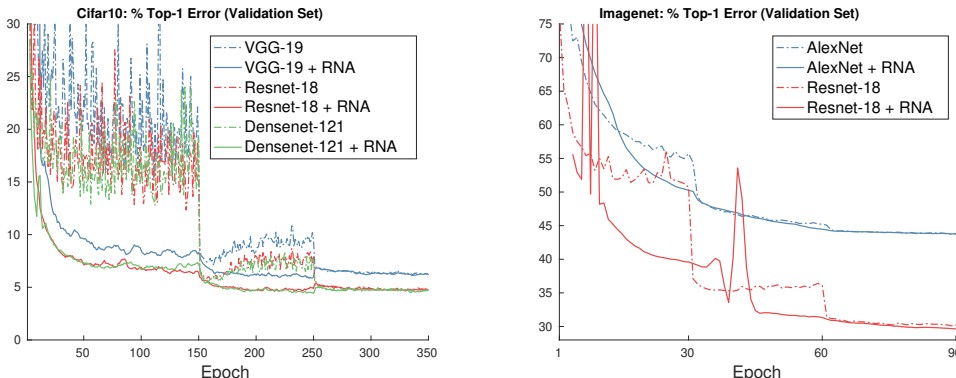

Figure 1: Comparison of Top-1 error between vanilla and extrapolated network.

## 3.1 CLASSIFICATION PIPELINES

Because the RNA algorithm is generic, it can be easily applied to many different existing CNN training codes. We used our method with various CNNs on CIFAR10 and ImageNet; the first dataset consists of $50k$ RGB images of size $32 \times 32$ whereas the latter is more challenging with $1.2M$ images of size $224 \times 224$. Data augmentation via random translation is applied. In both cases, we trained our CNN via SGD with momentum 0.9 and a weight decay of $10^{-5}$, until convergence. The initial learning rate is 0.1 (0.01 for VGG and AlexNet), and is decreased by 10 at epoch 150, 250 and 30, 60, 90 respectively for CIFAR and ImageNet. For ImageNet, we used AlexNet (Krizhevsky et al., 2012) and ResNet (He et al., 2016) because they are standard architectures in computer vision. For the CIFAR dataset, we used the standard VGG, ResNet and DenseNet (Huang et al., 2017). AlexNet is trained with drop-out (Srivastava et al., 2014) on its fully connected layers, whereas the others CNNs are trained with batch-normalization (Ioffe & Szegedy, 2015).

In these experiments, each $\theta_k$ corresponds to the parameters resulting from one pass on the data. We apply successively, off-line, at each epoch $j$ the RNA on $\{\theta_{j-K+1}, ...\theta_j\}$ and report the accuracy obtained by the extrapolated CNN on the validation set. Here, we fix $K = 10$ and $\lambda = 10^{-8}$.

## 3.2 NUMERICAL RESULTS

Figure 1 reports performance on the *validation set* of the vanilla and extrapolated CNN via RNA, at each epoch. Observe that RNA accuracy convergence is smoother than on the original CNNs which shows an effective variance reduction. In addition, we observe the impact of acceleration: the accelerated networks quickly present good generalization performance, even competitive with the best one. Note that several iterations after a learning rate drop are necessary to obtain acceleration because this corresponds to a brutal change in the optimization. Furthermore, selecting the hyper-parameter $\lambda$ can be tricky: for example, a larger $\lambda$ removes the outlier validation performance at epoch 40 of Figure 1, for ResNet-18 on ImageNet. Here, we have deliberately chosen to use generic parameters to make the comparison as fair as possible, but more sophisticated adaptive strategies has been discussed by Scieur et al. (2017).

Tables 3.2 reports the lowest validation error of the vanilla architectures compared to their extrapolated counterpart. Off-line optimization by RNA has only slightly improved final accuracy, but these improvements have been obtained *after* the training procedure, in an offline fashion, without extra-learning.

| Network | Vanilla | +RNA |
|---|---|---|
| VGG | 6.18% | 5.86% |
| Resnet18 | 4.71% | 4.64% |
| Densenet 121 | 4.50% | 4.42% |

| Network | Vanilla | +RNA |
|---|---|---|
| AlexNet | 43.72% | 43.72% |
| Resnet18 | 30.11% | 29.64% |

Table 1: Lowest Top-1 error on CIFAR10 (Left) and ImageNet (Right)

ACKNOWLEDGMENTS

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
