# OpenReview forum: "Nonlinear Acceleration of CNNs"
_ICLR.cc/2018/Workshop — Accept_

### Official Review · AnonReviewer2 · 2018-03-07
**The paper proposes to use the idea of regularized nonlinear acceleration to tune the performance of NN in an ad-hoc and offline manner.**

**Rating:** 7
**Confidence:** 3

**Review:**

Quality:

The paper brings forth an idea that can potentially improve accuracy using already fitted parameters. There is a slight concern regarding practicality of solving $(R^T R + \lambda I) z = 1$ when the number of parameters is extremely large (in millions) as is the case for most applications of CNN. However, since RNA procedure is applied off-line and in an ad hoc manner, there really is no reason to not try it. Informally speaking, if it works and leads to improved accuracy, then that's great. If it doesn't work, then since the fraction of time and effort needed by the user to configure RNA is relatively small compared to the time to train CNNs, it would not have been much of a trouble for the user.

Clarity:

The paper is clearly written.

Originality:

It is an application of the authors' previous work to image classification problems involving CNN.

Significance:

The experimental results are not ground breaking but it shows that it can be a useful method for fine tuning the performance of NN.

---

> ### Public Comment · (anonymous) · 2018-03-22
> **Thank you for your review**
>
> Dear Reviewer,
>
> thank you for your feedback. In fact, solving (R^T R+lambda I)z = 1 is extremely quick and easy. The size of the system is independent of the number of parameters. In fact, it grows in K^2 where K is the epoch counter, bounded by 10 in our experiments. The "difficult part" is the matrix multiplication R^T * R, which takes O(K^2 d) operations¨, where d is the number of parameters. In practice, this adds a negligible computation time, so we can easily compute the extrapolated model in parrallel on a CPU, without slowing down the learning algorithm.

---

### Official Review · AnonReviewer3 · 2018-03-09
**This paper is well written and the idea is interesting.**

**Rating:** 9
**Confidence:** 5

**Review:**

This paper presents an interesting extension of Regularized Nonlinear Acceleration (RNA)  for training deep CNNs. The paper is well written and easy to follow. The results are promising with clear explanations. The authors also publish their source code.

One minor question: There is a big jump of the proposed method in the right figure of Figure 2. Any reasons?

---

> ### Public Comment · (anonymous) · 2018-03-22
> **Thank you for your review**
>
> Dear Reviewer,
>
> thank you for your remark. The big jump in the figure 2 is due to our choice to use generic parameters to make the comparison as fair as possible. The spike is due to a low value of $\lambda$, and disapears if we use an adaptive strategy.

---

### Official Review · AnonReviewer1 · 2018-03-14

**Rating:** 9
**Confidence:** 5

**Review:**

This paper applies regularized nonlinear acceleration (RNA) to accelerate the training of of CNN. I find the empirical performance impressive, and believe this method can be adopted in practice. The authors also provide high-quality codes for the work.

---

> ### Public Comment · (anonymous) · 2018-03-22
> **Thank you for your review**
>
> Dear Reviewer,
>
> thank you for your positive feedback. We are still working on this project, and we also hope that other researchers will use this technique in the future.

---

### Public Comment · (anonymous) · 2018-03-22
**Public Code**

The code can be downloaded here:
https://github.com/windows7lover/RegularizedNonlinearAcceleration

This includes:
- The implementation of the RNA algorithm
- A tutorial code which shows how to use the algorithm (on mnist)
- The experimental code which reproduces the experiments on CIFAR10

---

### Decision · Program_Chairs · 2018-03-20
**ICLR 2018 Workshop Acceptance Decision**

**Decision:**

Accept

**Comment:**

Congratulations, your paper was accepted to the ICLR workshop.